# Over Two-Fold Photoluminescence Enhancement from Single-Walled Carbon Nanotubes Induced by Oxygen Doping

**DOI:** 10.3390/nano13091561

**Published:** 2023-05-06

**Authors:** Timofei Eremin, Valentina Eremina, Yuri Svirko, Petr Obraztsov

**Affiliations:** Department of Physics and Mathematics, Center for Photonics Sciences, University of Eastern Finland, Yliopistokatu 2, 80101 Joensuu, Finland; timofei.eremin@uef.fi (T.E.);

**Keywords:** photoluminescence, carbon nanotubes, doping, quantum defects

## Abstract

Covalent functionalization of single-walled carbon nanotubes (SWCNTs) is a promising way to improve their photoluminescent (PL) brightness and thus make them applicable as a base material for infrared light emitters. We report as high as over two-fold enhancement of the SWCNT PL brightness by using oxygen doping via the UV photodissociation of hypochlorite ions. By analyzing the temporal evolution of the PL and Raman spectra of SWCNTs in the course of the doping process, we conclude that the enhancement of SWCNTs PL brightness depends on the homogeneity of induced quantum defects distribution over the SWCNT surface.

## 1. Introduction

Single-walled carbon nanotubes (SWCNTs) are hollow cylindrical carbon structures with a nanometer-scale diameter and length ranging from a few tens of nanometers to tens of microns [1]. One of the most intriguing properties of SWCNTs is their stable excitonic near-infrared photoluminescence (PL), which depends on their diameter and chirality [2]. This has generated great interest in the development of SWCNT-based infrared light emitters, including single-photon sources [3], LEDs, and bioimaging probes [4]. However, a very low quantum yield [5], which is suppressed by exciton quenching on defective SWCNT sites [6] due to free exciton diffusion, essentially limits the application range of SWCNTs light emitters in practice. Another critical factor that suppresses the PL quantum yield is the presence of dark excitons. Being energetically downshifted compared with bright excitons, they provide an additional pathway for the nonradiative relaxation of photoexcitation [7].

These limitations can be overcome by introducing traps along the SWCNT by covalent sp^3^ functionalization, which is a promising way to improve the PL brightness. Specifically, this can be completed by adding functional groups capable to create “bright” sp^3^ defects, in the vicinity of which the trapped excitons will radiatively recombine before quenching on intrinsic SWCNT defects [8,9,10,11,12,13,14,15]. The energy level of such a trapped exciton might be energetically downshifted compared to the dark exciton level thus preventing the quantum yield suppression. Eventually, such sp^3^ functionalization of SWCNTs may lead to an improvement in the PL brightness of the SWCNTs by a factor of 4.2 [16]. Furthermore, doping SWCNT with sp^3^ not only results in a substantial enhancement in the PL brightness but also induces photon antibunching, enabling the use of sp^3^doped SWCNT as single-photon emitters even at room temperature [17,18]. This phenomenon can be explained by the localization of excitons near artificially induced defects, where the trapping potentials are deep enough to prevent thermal de-trapping even at room temperature. Nevertheless, it should be noted that the sp^3^ doping process for SWCNT is complex and time-consuming, involving multiple steps that may take from several hours [14] to several days [16].

Oxygen doping provides an alternative method for creating zero-dimensional traps in SWCNTs. Oxygen adducts in oxygen-doped carbon nanotubes can take on several chemical configurations, including the formation of an epoxide bond between an oxygen atom and two adjacent sp^3^ carbon atoms and the formation of an ether bond between an oxygen atom and two sp^2^ carbon atoms nearly parallel (ether-l) or perpendicular (ether-d) to the nanotube axis through C-O-C bonds [19]. The most stable configuration, ether-d, is primarily responsible for the changes observed in the optical properties of the SWCNTs resulting from oxygen doping [20]. Oxygen defects in SWCNT structure also enable deep trapping of excitons providing efficient single-photon emission [17] and also possess good perspectives for spectrally-resolved bioimaging [4]. The conventional oxygen doping technique is also quite a time-consuming procedure, which involves the exposure of SWCNT to ozone with subsequent UV irradiation of up to 16 h [4,20,21,22,23,24]. However, it has been recently demonstrated [25] that boosting PL brightness by a factor of 1.36 on a minute timescale can be achieved by a very simple method of SWCNT oxygen doping based on the photodissociation of hypochlorite ions. The technological simplicity, efficiency, and scalability of this method make it highly promising for the development of IR light emitters based on oxygen-doped SWCNTs (O-SWCNTs) provided a higher brightness enhancement factor will be achieved.

Here, we report a modification of the oxygen doping method based on hypochlorite ion photodissociation, which provides a record high—by more than a factor of 2—enhancement of the SWCNT PL brightness. By analyzing the defect formation in the structure of SWCNTs during oxygen doping, we revealed that the homogeneity of the oxygen atom distribution along the SWCNT surface is of crucial importance for achieving the bright PL of oxygen-doped SWCNTs.

## 2. Materials and Methods

The commercially available (6,5)-enriched CoMoCat SWCNT purchased from Merk was suspended in a concentration of 0.1 mg/mL in Sodium dodecyl sulfate (SDS) solution (2%) by 4-h tip-sonication (model Hielsher UP200, manufacturer Hielscher Ultrasonics GmbH (Teltow, Germany), integrated power 200 W, on/off cycle ratio 0.1, volume of the treated suspension 50 mL), followed by 1 h ultracentrifugation (125,000× *g*). Approximately 80% of the supernatant was collected and used for further research. Before doping, the SWCNT suspension was diluted 10 times with water to reduce the SDS concentration to 0.2%.

The employed doping procedure is a modification of the method reported in [25]. NaOCl solution (2 mg/L) was added to the SWCNT suspension at a concentration of 50 µL/mL. Subsequently, an open quartz cuvette (10 × 10 × 35 mm) with 2 mL of SWCNT suspension was centered right under the UVC mercury lamp (model UVL-10 Ozone, manufacturer LLC “UVL” (Istra, Russia), max intensity at 254 nm, power density 27 µW/cm^2^). The distance between the lamp and the surface of the suspension was approximately 15 mm. Doping was performed by exposing the suspension to UV light for a certain period of time. This method allowed us to temporarily suspend doping and perform spectroscopic measurements.

To perform Raman measurements, the cuvette was inclined and the sample was excited through the cuvette wall with a laser beam focused 2–3 mm below the cuvette edge. A diode laser with a wavelength of 532 nm was utilized as the excitation source, and the signal was collected in a backscattering configuration.

The suspensions were subjected to spectroscopic measurements using the same cuvette used for the doping. PL maps were obtained by exciting the samples with a Xe lamp for visible range excitation and a tunable CW titanium-sapphire laser for NIR excitation. The reported PL spectra were acquired under excitation with an 840 nm wavelength Ti:sapphire laser (model 3900s, manufacturer Spectra-Physics, Milpitas, CA, USA), unless stated otherwise. The excitation beam was defocused on the sample (diameter of the beam was 2–3 mm) and the PL signal was detected using an InGaAs array.

A two-channel spectrophotometer (model Lambda 900, manufacturer Perkin Elmer, Waltham, MA, USA) was used to measure the optical absorption spectra. The thickness of the optical path was 10 mm.

## 3. Results

Figure 1a displays the excitation-emission PL map of a pristine SWCNT suspension. The color scale is normalized based on the maximum intensities for the 475–720 nm and 720–1000 nm spectral regions, separately. This normalization was required because different excitation sources, the Xe lamp and CW Ti:sapphire laser, were used for these regions with varying power levels. The main observed PL features are indicated with notation $(n,m{)}_{abs}^{em}$, where (n,m) refers to the chiral indices of the corresponding SWCNT, which describe the crystalline structure of the SWCNT, including diameter and chirality, and thus the energy of excitonic transitions. The subscript and superscript indices describe absorptive and emissive energy levels during the PL process, respectively. Therefore, the predominant spectral feature $(6,5{)}_{22}^{11}$ corresponds to light absorption by the E_22_ exciton of (6,5)-SWCNT and subsequent light emission by the E_11_ exciton of SWCNT with (6,5) chirality.

Other optical features correspond to residual PL of SWCNTs having (6,4), (8,3), (7,5), (8,4), and (7,3) chiralities. The assignment of the optical features to SWCNT chiralities is based on the dataset published in [2].

The diameters of SWCNTs in the suspension range from 0.69 nm to 0.84 nm. The length of SWCNTs cannot be accurately estimated using the employed spectroscopic methods, but it is expected to be on the order of a few hundred nanometers based on the employed manufacturing of the samples [26].

For purposes of the further discussion we note here the observation of emission and absorption phonon-side bands (PSB), which result from the involvement of a phonon in the process of exciton recombination and excitation, respectively. For example, the $(6,5{)}_{11PSBabs}^{11}$ spectral feature with a resonant excitation wavelength of 840 nm and an emission wavelength of 980 nm, corresponds to the simultaneous excitation of a K-momentum dark exciton from the E_11_ manifold and the band-edge phonon with subsequent light emission by bright E_11_ exciton [5]. Taking into account spectral positions of the $(6,5{)}_{11PSBabs}^{11}$ and $(6,5{)}_{11}^{11PSBem}$ phonon-side bands to determine the position of the ($6,5{)}_{11}^{11}$ spectral feature which is marked with a dashed circle. This is not observable in the performed experiment due to huge scattering of laser excitation line (brown rectangles in Figure 1a,b)

Upon doping the SWCNTs with NaOCl and exposing them to UV light, significant changes in the PL map were observed, as shown in Figure 1b. Specifically, new emission bands appeared at approximately 1145 nm and 1130 nm, which are indicated by dashed vertical lines, whereas, the original spectral features were partially suppressed (see Appendix A in the Appendix A for the comparison of peak intensities).The new spectral feature with an emission wavelength centered at 1130 nm has several resonant excitation wavelengths, namely, 568 nm, 840 nm, and 980 nm, which perfectly match the resonance conditions of $(6,5{)}_{22}^{11}$, $(6,5{)}_{11PSBabs}^{11}$, and $(6,5{)}_{11}^{11}$, respectively (see Figure 1b). Based on this observation, we attribute the emission band at 1130 nm to the emission from (6,5)-SWCNT. Similarly, the emission line centered at 1045 nm should be associated with the (6,4)-SWCNT. No new emission bands associated with SWCNTs of other chiralities were observed because of their low concentration in the sample.

The emission wavelengths of the new spectral features are close to previously reported values for the recombination energy of excitons localized in the vicinity of O atoms in oxygen-doped carbon nanotubes [20]. This confirms efficient oxygen doping of SWCNTs via interaction with NaOCl under UV exposure and allows the determination of the physical nature of the new emission bands as a result of the radiative recombination of these zero-dimensional excitons. The localized exciton energy level is marked with a ‘*’ sign.

Figure 1c shows the evolution of the SWCNTs PL spectrum during the doping procedure. To focus on SWCNT with predominant (6,5) chirality, we used an excitation wavelength of 840 nm, which coincides with the absorbance PSB resonance of (6,5) SWCNT and is spectrally well-separated from the absorption resonances of SWCNT with other chiralities. Further, we omit the detailed notation of the PL spectral features and use a short designation (6,5) instead of $(6,5{)}_{11PSBabs}^{11}$ and (6,5)* instead of $(6,5{)}_{11PSBabs}^{*11}$. As the UV exposure time increased, the intensity of the (6,5) band decreased monotonically.The behavior of the (6,5)* band in the PL spectrum is more complicated; the intensity of the new band increases until it reaches its maximum after approximately 20–25 min of UV exposure. The subsequent UV exposure led to a decrease in the intensity of the (6,5)* band.

Figure 1d shows the PL spectrum of the doped SWCNT with the maximum PL signal (25 min of UV exposure, red line), along with the PL spectrum of the pristine SWCNT (blue line). The maximum peak intensity of the (6,5)* band in the PL spectrum of the oxygen-doped SWCNT is more than two times higher than that of the (6,5) band in the PL spectrum of the pristine SWCNT. It should be noted that we performed a control experiment in which SWCNT were irradiated by UV light under the same conditions, but without NaOCl (deionized water instead of NaOCl solution was added to the SWCNT suspension in the same volume), and we did not observe any signs of the (6,5)* and (6,4)* peaks in the PL spectrum. (yellow reference line in Figure 1d). Further, there were no significant changes in the PL spectrum of SWCNTs after 10 h of interaction with NaOCl at the same concentration without UV exposure. This confirms that both NaOCl and UV light are necessary for effective oxygen doping of the SWCNT.

To additionally characterize the doping reaction and track any possible changes in the morphology of SWCNT in the suspension, we performed UV-vis-NIR optical absorption spectroscopy measurements. Figure 2 compares the optical absorption spectra of the SWCNT suspension before and after doping for different reaction durations. It is seen that both E11 optical transitions in the spectral region of 800–1200 nm and E22 optical transitions in the spectral region 400–800 nm are suppressed though changes in the E11 spectral region are more prominent. A similar effect was previously reported for SWCNT doped with other methods and is commonly explained in terms of the suppression of the exciton oscillator strength due to the depletion of electron states in the valence band of SWCNT [27,28]. In all other aspects, the optical absorption spectrum remains almost unchanged, suggesting the absence of any drastic changes in the SWCNT morphology. Moreover, no changes in the background spectral shape related to scattering indicate a similar suspendability for the pristine and oxygen-doped SWCNT.

Now we proceed to the comparison of the achieved PL enhancement with previously published results. The most physically meaningful way to do so is to consider the relative changes in PL quantum yield which might be calculated as:(1)$$\frac{QY(doped)}{QY(pristine)}=\frac{I_{doped}^{\left(6,5*\right)}}{I_{pristine}^{\left(6,5\right)}}$$
where $I_{doped}^{\left(6,5*\right)}{\left(6,5\right)}^{*}$—is the integrated intensity of the (6,5)* peak in the spectrum of doped SWCNT and $I_{pristine}^{\left(6,5\right)}\left(6,5\right)$—is the integrated intensity of the (6,5) peak in the spectrum of *pristine* SWCNT. Applying spectra deconvolution to separate PL peaks and integration (see Appendix A), we estimated that oxygen doping enhances the PL QY by at least 6 ± 0.6 times. Note that this is a lower-bound estimation because formula (1) is accurate for the case of the same optical density of the sample at the excitation wavelength. In fact, the optical density of the sample at 840 nm is slightly decreased due to oxygen doping (see Figure 1d) meaning that the real enhancement factor of the PL QY is higher than 6. However, a direct comparison of this value with the results published in the literature is not possible because of the lack of published data on spectral deconvolution and integrated peak intensity values.

Instead, we define the PL brightening coefficient ϑ as:(2)$$\vartheta =\frac{A_{doped}^{(6,5)*}}{A_{pristine}^{\left(6,5\right)}}$$
where $A_{doped}^{\left(6,5*\right)}$ is the amplitude (maximum intensity) of the (6,5)* band in the PL spectrum of O-doped SWCNTs and $A_{pristine}^{\left(6,5\right)}$ is the amplitude of the (6,5) band in the PL spectrum of pristine SWCNT (before doping). The higher ϑ value means the higher $\frac{QY(doped)}{QY(pristine)}$ value, although from a practical point of view, such as the application of O-SWCNTs as IR light emitters, the comparison of ϑ values is more meaningful since it reflects changes in PL brightness under the same excitation conditions. Furthermore, the previously published literature may provide accurate ϑ values that can be extracted from PL spectra.

In this work, we achieved a highly reproducible ϑ value of 2.12 ± 0.11 (Figure 1d). Table 1 compares the ϑ values among all of the published literature on the oxygen doping of SWCNT. As shown in Table 1, our method demonstrates the highest PL emission enhancement.

## 4. Discussion

Let us now discuss the reason for such an advanced PL intensity enhancement observed in this work in comparison to the previously published results. Note that the SWCNTs used in this study are similar to those used in most of the papers listed in Table 1, indicating that the reported variance in θ values is mainly caused by the details of the oxygen doping procedure.

The higher *ϑ* values obtained in this work compared to the previously published results obtained in [4,20,21,22,23,24,29] can easily be ascribed to the principal differences in the employed chemical reactions. Indeed, oxygen doping of SWCNT via ozonization occurs in several steps, including the formation of an ozonide adduct on the surface of SWCNT, subsequent loss of O_2_ with the formation of an epoxide adduct, and photoisomerization of the epoxide adduct into either one [20]. Each stage of this reaction occurs on the surface of the SWCNT and may eventually lead not only to the formation of bright excitonic states but also to the formation of dark trapping defects, thus preventing the conversion of diffusive (6,5) excitons into localized (6,5)* and limiting the resulting PL brightness [13]. In contrast, in the NaOCl-based approach, the formation of oxygen defects on the surface of SWCNT occurs in a single-step reaction of carbon walls with atomic oxygen previously produced at a distance from the SWCNT surface during photodissociation of the hypochlorite ion. This difference in reaction mechanisms explains the higher ratio of bright and dark defects in the case of using NaOCl as an oxygen source, and therefore, higher PL brightening coefficients.

In our study, we used the same doping reaction as in [25], which involves the photodissociation of hypochlorite ions. However, we found that the resulting *ϑ* values obtained in this work and in [25] vary significantly. To determine the reason for this difference, we utilized Raman spectroscopy to investigate the details of the defect formation during oxygen doping.

The evolution of the Raman spectra is shown in Figure 3a. A gradual decrease in the tangential G-mode intensity was observed during the doping reaction, which is consistent with the changes in the intensity of the (6,5) PL band (Figure 3b). The intensity of the radial breathing mode (RBM) in the Raman spectra also monotonically decreased with increasing UV exposure duration (See Appendix A). Simultaneous suppression of the RBM and G modes in the Raman spectrum can be considered as a fingerprint of the effective doping of SWCNT [30,31,32]. The absence of other drastic changes in the RBM spectral region of the Raman spectrum during oxygen doping suggests that the morphology of the SWCNT in the sample remains unchanged. Based on the Raman shift of the RBM mode (275 cm^−1^), we estimated the diameter of the investigated SWCNT as 0.86 nm which is consistent with the data obtained using PL mapping.

Information about defects formation in SWCNT can be obtained by analyzing the changes in the spectral region of the so-called “defective” (D) mode (inset in Figure 3a). The intensity of the D-mode increased during the first 50 min of the reaction and then began to decrease, resembling the (6,5)* PL peak behavior (Figure 3c). The density of defects in SWCNT can be estimated from the ratio of the integrated intensities of the D and G bands using the calibration curve reported in [33]. Thus, for moderately doped SWCNT (25 min of UV exposure), the density of oxygen defects is estimated as 30 µm^−1^.

This observation confirms that the (6,5)* PL peak appears owing to defect formation by the covalent bonding of oxygen atoms to the SWCNTs walls. However, when the UV exposure time exceeded 25 min, the intensity of the (6,5)* peak in the PL spectrum began to decrease, although the intensity of the D-mode continued to increase. This cannot be explained by the amorphization of the SWCNTs crystalline structure due to a large number of oxygen defects because the shapes of the D and G bands do not demonstrate any signs of amorphization until 50 min of UV exposure. Indeed the disruption of the SWCNTs structure progresses only after 50 min of UV illumination, which is accompanied by an abrupt drop in both the D and G band intensities, a sharp increase in the D/G intensity ratio, and distinctive changes in the shape of the D-band in the Raman spectrum (violet line in Figure 3a).

To exclude the suppression of the (6,5)* PL peak due to consumption of NaOCl reactant during the first 25 min [25] we performed addition measurement. By adding an extra portion of non-irradiated NaOCl to the SWCNTs suspension in the middle of the O-doping process (30 min of UV irradiation), we did not reverse the downward trend of the (6,5)* peak intensities versus time of UV irradiation (see Appendix A). This result declines a scarcity of NaOCl reactant as a limiting factor of the (6,5)* peak growth.

We propose that the proximity of O-defects might interfere with the localizing potential and electronic structure of O-doped sites, which may prevent effective radiative emission and thus limit the growth of the (6,5)* peak intensity despite the increase in the total number of oxygen defects. This means that the homogeneity of doping is a critical issue for achieving bright emissions from the O-doped sites of SWCNTs.

This assumption can explain the higher PL brightening coefficient ϑ achieved in this work compared with previously published results. Indeed, the use of prolonged (25 min) and gentle (27 µW/cm^2^) UV exposure in this work should provide more uniform O-doping than in the case of brief (40 s) and rough (29 mW/cm^2^) UV exposure, which was applied in [25] and resulted in a striking difference in ϑ value (2.12 in this work and 1.36 in [25] with almost the same D/G intensity ratio).

Employing SWCNT doped with oxygen using NaOCl and prolonged and gentle UV exposure as bioimaging probes may provide better contrast in tissue images than those obtained using conventional methods. In the context of single-photon sources, this approach may provide a higher emission frequency. Nevertheless, for successful industrial applications of O-SWCNT IR light emitters, including bioimaging probes and single-photon emitters, as well as IT LED and IR nanolasers, the brightness of O-SWCNT PL should be further increased. The results obtained in this work suggest an efficient way to do so by adjusting the duration and intensity of UV exposure, thus improving the homogeneity of the SWCNTs O-doping. This approach is a promising way to overcome the brightness of sp_3_-doped SWCNTs while maintaining the simplicity and rapidity of O-doping technology.

## 5. Conclusions

The demonstrated modification of the SWCNT oxygen doping method results in the appearance of a red-shifted PL band with a peak intensity that is more than twice that of the peak intensity of the PL band in the undoped sample. The PL brightening coefficient of 2.12 ± 0.11 obtained in this work is the highest value among all published works using the O-doping approach. The Raman spectroscopy study of defect formation suggests that this is achieved owing to a more homogeneous distribution of oxygen atoms along the SWCNTs due to prolonged and gentle UV exposure. We propose that further optimization of the O-doping method by adjusting the duration and intensity of UV exposure is a promising way to increase the brightness of O-SWCNT photoluminescence while maintaining the other advantages of this approach, such as technological simplicity and rapidity, which satisfy the requirements of technological applications such as IR LEDs, IR nanolasers, IR single-photon sources, and bioimaging probes.

## Figures and Tables

**Figure 1 nanomaterials-13-01561-f001:**
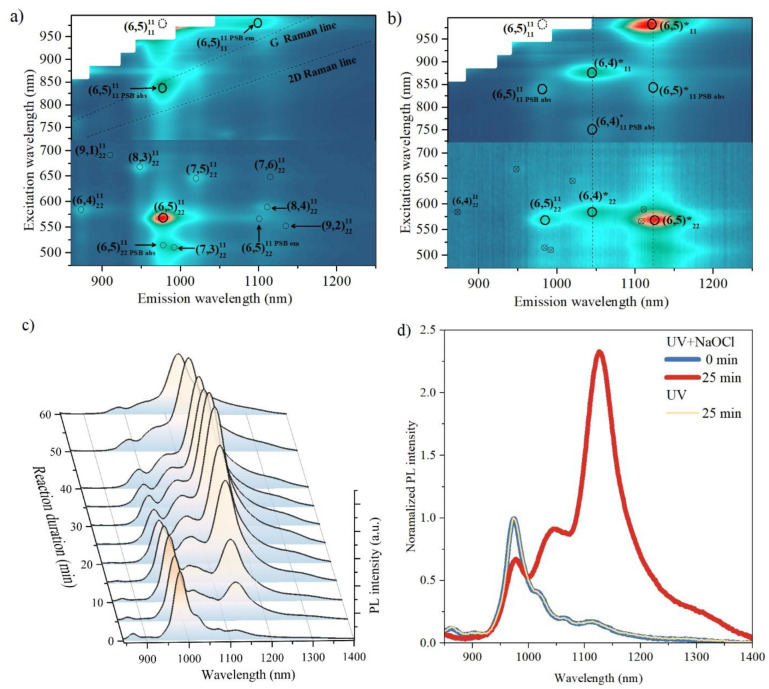
Excitation-emission photoluminescence (PL) maps of pristine single-walled carbon nanotubes (SWCNTs) (**a**) and O-doped SWCNTs (**b**). A contrast horizontal lines at 720 nm separate maps obtained using different excitation source (Xe lamp for visible and Ti:sapphire laser for NIR), color scale normalized separately for each part (blue for 0 and red for 1). New spectral features appeared due to the oxygen doping are marked with * sign. (**c**) Evolution of PL spectra of SWCNTs during the doping procedure. (**d**) Comparison of SWCNT PL spectra before O-doping and at the optimum doping concentration. The excitation wavelength is 840 nm.

**Figure 2 nanomaterials-13-01561-f002:**
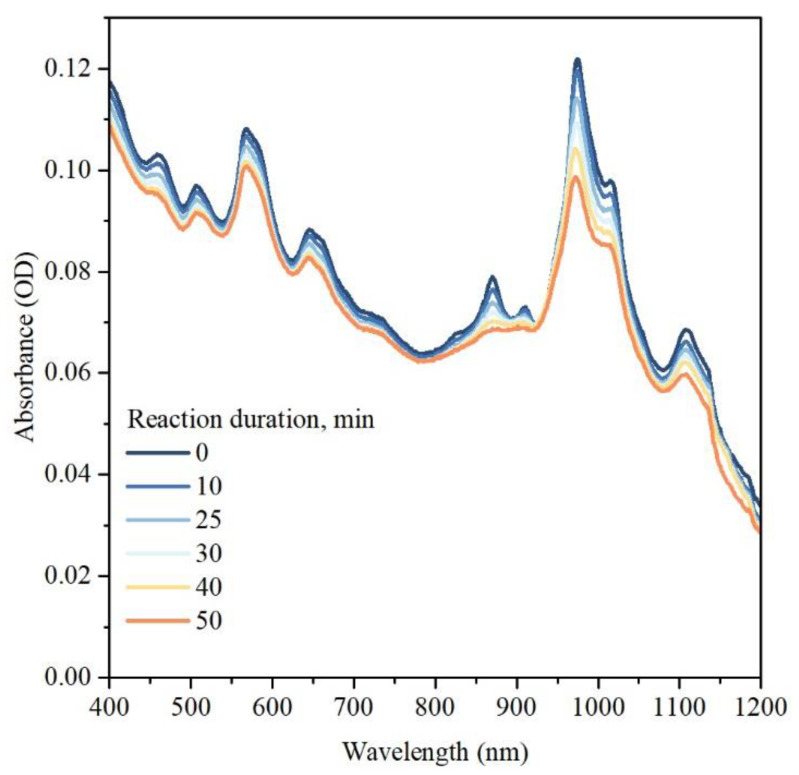
UV-vis-NIR optical absorption spectra of pristine and doped SWCNT.

**Figure 3 nanomaterials-13-01561-f003:**
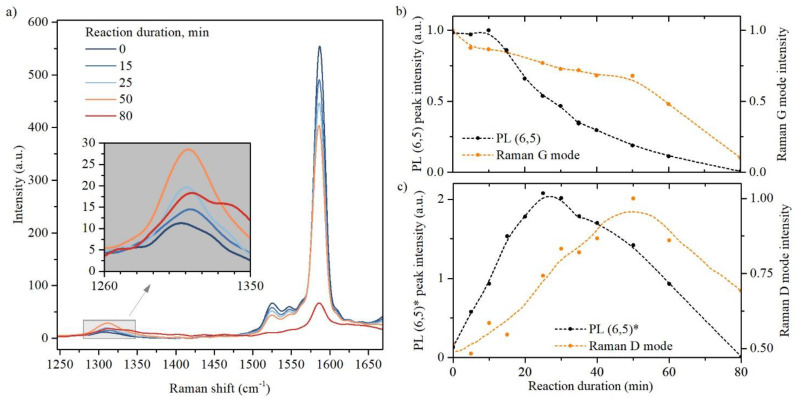
(**a**) Raman spectra of pristine and oxygen-doped SWCNTs; (**b**) the dependence of the PL (6,5) and Raman G-mode intensity on the duration of the doping reaction; (**c**) the dependence of the PL (6,5)* and Raman D-mode intensity on the duration of the doping reaction. Dashed lines are a guide-to-the-eye.

**Table 1 nanomaterials-13-01561-t001:** A comparison of the photoluminescence (PL) brightening coefficient ϑ achieved employing oxygen doping of single-walled carbon nanotubes (SWCNT).

Ref.	PL Brightening Coefficient ϑ	Chirality	Synthesis Method and Dealer	Note on O-Doping Method
This work	2.12	(6,5)	CoMoCat, Merk (former Sigma-Aldrich, St. Louis, MO, USA)	NaOCl + light
(Chiu et al.) [29]	1.65	(6,5)	CoMoCat, Sigma-Aldrich	treatment with polyunsaturated fatty acids
(Lin et al.) [25]	1.35	(6,5)	CoMoCat, Sigma-Aldrich	NaOCl + light
(Miyauchi et al.) [21]	0.86	(6,5)	CoMoCat, dealer unknown	O_3_ + light
(Ghosh et al.) [20]	0.68	(6,5)	unknown	O_3_ + light
(Iwamura et al.) [22]	0.61	(6,5)	CoMoCat, Sigma-Aldrich and Southwest Nanotechnologies (Norman, OK, USA)	O_3_ + light
(Iizumi et al.) [4]	0.51	(6,5)	unknown	O_3_ + light
(Akizuki et al.) [23]	0.33	(6,5)	CoMoCat, Southwest Nanotechnologies	O_3_ + light

## Data Availability

Publicly available datasets were analyzed in this study. This data can be found here: https://drive.google.com/file/d/156TeiFf7ahVhYjm__CdkiKDSUIpEtHdg/view?usp=sharing (accessed on 3 May 2023).

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
