# Peer review of "Over Two-Fold Photoluminescence Enhancement from Single-Walled Carbon Nanotubes Induced by Oxygen Doping"

_nanomaterials, 2023, doi:10.3390/nano13091561_

Round 1

Reviewer 1 Report

In this paper, the authors used an oxidation method to introduce oxygen dopants intro SWNT structure. PL spectra illustrated a more than 2 times increase in the PL emission under the same test conditions when SWNTs were oxidized under optimized conditions. The influence of reaction time was studied using PL spectra and the D/G ratio of Raman spectra. Overall, this is an interesting paper but more analysis and evidence are required to illustrate the mechanism for this change and open doors to other related optical and catalytic areas. Thus, I suggest a major revision before publishing it.  

1.       Lines 36-37, Page 1. It was claimed that “Alternatively, a zero-dimension trap can be created by the oxygen doping of SWNTs, i.e. by introducing oxygen bridge between carbon atoms of the SWNT surface. ” However, the difference between oxygen doping and traditional sp3-doped is not clearly stated. The carbon atoms binding to oxygen could also be sp3-type.   

2.       Lines 57-58, Page 2. “After that, it was irradiated by UVC light (max at 254 nm) from a mercury lamp (27 μW/cm2) for a certain time.”The methods part should be more detailed so that the results could be repeated by the readers. For example, where is the light source located? Is it in the suspension or on the top of the suspension? This could cause huge differences in light absorption, temperature, and other conditions.

3.       What is the size of the SWNT? Do morphology and size change before and after the UV treatment? Does the SWNT suspend better after the treatment? More characterizations are needed.

4.       Discussion part, Pages 4-5. How does the concentration of defects change after the treatment? What is the average distance between defects? Are they isolated or not? Detailed analysis of Raman spectra for the changes is necessary. In addition, the RBM peak is also important for SWNT. How does that peak evolve? I believe this could give more information about the crystalline structure changes.

5.       What are the quantum yields of SWNTs and oxidized SWNTs? Compared to the benchmark, are these numbers close to practical use?

Not bad.

Reviewer 2 Report

Honestly, this manuscript is not in my personal top among recent papers I have seen. My initial feeling has been as it is not very important report bringing just incremental knowledge. However, upon certain thinking over, I feel like this manuscript is maybe not so comprehensive as the broad topic suggests, but is on the other hand very clear and well focused. I think, it can be published after addressing some minor comments listed below. 

1. I would suggest to use more common abbreviation for single-walled carbon nanotubes (SWCNT) instead of SWNT used by the authors.

2. The authors have claimed the achieved enhancement in the PL brightness (by about 2.1 times) to be "record" (lines 46-48) and "outstanding" (line 124) in comparison with ~1.6 times [25] and ~1.4 times [24].

Is this difference indeed so important in practice? Of course, I understand that 2.1 is better than 1.6, but I cannot pretend a real-life situation when the 1.6x enhancement is not acceptable but 2.1x enhancement resolves some practical issue.

3. Moreover, comparing the "brightening coefficients" (in the authors' terms), it is important to keep in mind that the value of the achieved enhancement is strongly dependent on the initial state. How can one be sure that the pristine CNTs used in refs. 3, 19-22, 24, 25 were of the same brand, purity, and quality? I suggest extending the discussion of Fig. 2 by adding more details about other references studies, not only the best 24 and 25. Probably, organizing this comparison in a table can be advantageous, since some additional data (nanotubes source, treatment procedure, excitation conditions, etc) could be collected in a compact shape.

4. In line 55 it is stated that "SWNT suspension was diluted 10 times with water". However, initial concentration (or volume) of the suspension and the mass of the nanotubes are not mentioned. I would also add the power of the tip sonication and its mode (I do not think it was continuous).

5. Please explain how the assignments of the bands to the nanotubes types (lines 65-67) were made. 

6. In line 70 it is written that "the original spectral features were suppressed" upon the treatment. However, it is not clear whether the maps in Figs. 1A, B display the absolute spectral intensity, or each map was normalized independently. In other words, it is indeed true that the original bands were weakened - or the new band was just much stronger that the original ones?

7. In lines 82-89 it is not completely clear to me why the kinetics of the emission spectrum evolution was tracked using excitation at a wavelength, data for which is absent in Figs. 1A,B. Moreover, from the overall shape of the maps in Figs. 1A,B it is not so evident that excitation at 840 nm should have induced emission at all.

8. Comparison of the QY as directly proportional to the emission intensity (line 93) is valid only if the sample absorbance is not changed upon the treatment. Do the authors have the corresponding data?

9. Regarding the general conclusion that longer UV treatment under mild conditions is advantageous over 'flash' illumination, do the authors have any idea on the possibility of another extreme control case (several days treatment with NaOCl in dark)?

10. Are the nanotubes with the so introduced defects stable during storage? What about comparison of the pristine and the treated nanotubes considering their stability against photobleaching? 

Please reconsider the language throughout the entire manuscript. The mistakes I have noticed are not awful, but they are too numerous to be acceptable. Let me give just several examples (not the complete list) from the first paragraphs.

Line 21. 'Of' is possibly missing in "application range of(?) SWNT in practice".

Line 25. 'Overcome' should be instead of 'overcomed'.

Line 29. "on an intrinsic SWNT defects" - either 'an' is obsolete or 'defect' should be in Single.

Line 33. 'That' is likely obsolete.

I suggest the authors to ask a native English-speaking colleague to perform a quick proofreading of the manuscript.

Round 2

Reviewer 1 Report

I believe the authors have answered all my questions and thus, I suggest publishing it as it is.